# Comparative Phosphoproteomic Analysis of Sporulated Oocysts and Tachyzoites of *Toxoplasma gondii* Reveals Stage-Specific Patterns

**DOI:** 10.3390/molecules27031022

**Published:** 2022-02-02

**Authors:** Ze-Xiang Wang, Liang Che, Rui-Si Hu, Xiao-Lin Sun

**Affiliations:** 1College of Veterinary Medicine, Gansu Agricultural University, Lanzhou 730070, China; cl2855546074@163.com (L.C.); sunxl@gsau.edu.cn (X.-L.S.); 2Lanzhou Veterinary Research Institute, Chinese Academy of Agricultural Sciences, Lanzhou 730046, China; grishu0707@gmail.com

**Keywords:** *Toxoplasma gondii*, life cycle, IBT, phosphoproteomics

## Abstract

*Toxoplasma gondii* is an obligate intracellular protozoan of severe threat to humans and livestock, whose life history harbors both gamic and apogamic stages. Chinese 1 (ToxoDB#9) was a preponderant genotype epidemic in food-derived animals and humans in China, with a different pathogenesis from the strains from the other nations of the world. Posttranslational modifications (PTMs) of proteins were critical mediators of the biology, developmental transforms, and pathogenesis of protozoan parasites. The phosphoprotein profiling and the difference between the developmental phases of *T. gondii*, contributing to development and infectivity, remain unknown. A quantitative phosphoproteomic approach using IBT integrated with TiO_2_ affinity chromatography was applied to identify and analyze the difference in the phosphoproteomes between the sporulated oocysts and the tachyzoites of the virulent ToxoDB#9 (PYS) strain of *T. gondii*. A total of 4058 differential phosphopeptides, consisting of 2597 upregulated and 1461 downregulated phosphopeptides, were characterized between sporulated the oocysts and tachyzoites. Twenty-one motifs extracted from the upregulated phosphopeptides contained 19 serine motifs and 2 threonine motifs (GxxTP and TP), whereas 16 motifs identified from downregulated phosphopeptides included 13 serine motifs and 3 threonine motifs (KxxT, RxxT, and TP). Beyond the traditional kinases, some infrequent classes of kinases, including Ab1, EGFR, INSR, Jak, Src and Syk, were found to be corresponding to motifs from the upregulated and downregulated phosphopeptides. Remarkable functional properties of the differentially expressed phosphoproteins were discovered by GO analysis, KEGG pathway analysis, and STRING analysis. S8GFS8 (DNMT1-RFD domain-containing protein) and S8F5G5 (Histone kinase SNF1) were the two most connected peptides in the kinase-associated network. Out of these, phosphorylated modifications in histone kinase SNF1 have functioned in mitosis and interphase of *T. gondii*, as well as in the regulation of gene expression relevant to differentiation. Our study discovered a remarkable difference in the abundance of phosphopeptides between the sporulated oocysts and tachyzoites of the virulent ToxoDB#9 (PYS) strain of *T. gondii*, which may provide a new resource for understanding stage-specific differences in PTMs and may enhance the illustration of the regulatory mechanisms contributing to the development and infectivity of *T. gondii*.

## 1. Introduction

*Toxoplasma gondii* is a monocellular and intracellular protozoa that obligately subsists in host cells. It belongs to the Apicomplexa family that includes several protozoans of veterinary and medical importance, including pathogenic agents of the malaria *Plasmodium* species and the common cause of infant diarrheal sickness, *Cryptosporidium* [1]. The subpopulation structures of *T. gondii* were different in geographical regions due to its sparsely sexual recombination and population cleaning [2,3]. Thus, three major clonal lineages, which were referred to as subtypes I, II, and III, were found in North America and Europe in history [4]. Lately, most of the strains isolated from wildlife in North America were designated as type 12, the fourth subtype [5]. However, studies of the epidemiological distribution involving *T. gondii* isolated from food-derived animals and humans in China revealed Chinese 1 (ToxoDB#9) was the predominant genotype, and this genotype harbored two virulent factors, GRA15_II_ and ROP16_I/III_. The two virulent factors were diverse in the type I, II, and III strains of *T. gondii*. This suggested the pathogenesis of the Chinese 1 (ToxoDB#9) strains was different from the strains isolated from other regions of the world [6,7].

The life cycle of *T. gondii* has both a gamic stage and an apogamic stage, which exclusively occur in the definitive host and the extensive coverage of warm-blooded intermediate hosts, respectively [1]. The sexual phase takes place in the cat gut epithelium and ultimately generates oocysts. After being ingested by a cat, released bradyzoites of *T. gondii* invade the enterocytes, and coccidian development occurs, including several rounds of schizogony and gamogony, during which diploid zygotes are produced by the fertilization of microgametes and macrogametes [8,9], and an oocyst wall is formed through sequential secretion of wall-forming bodies type 1 and type 2 from mature macrogametes [10]. Subsequently, unsporulated oocysts are secreted in the feces of the cat and developed into infectious sporulated oocysts that host two sporocysts, each harboring four haploid sporozoites in an outside aerobic environment [10]. Oocysts exhibit resistance to varied harsh conditions and inactivation agents or procedures, for example, temperatures below 60 °C, strong acids, household bleach and detergents, chlorine dioxide and ozone applied in the water industry, and even ultraviolet irradiation in high doses (≥500 mJ/cm^2^) [10,11,12,13]. The bilayered oocyst and sporocyst wall composed of proteins was deemed to be the critical structure and protective mechanism that allowed *T. gondii*’s survival in harsh conditions despite the sporozoites possessing protective mechanisms to defeat some external stresses [10,14,15,16]. Oocysts not only play a key role in *T. gondii* infection of herbivores but are also a main source of omnivore infections. It was reported that 21% of waterborne toxoplasmosis outbreaks between 1976 and 2009 were derived from oocysts [10,17]. Disappointingly, the mechanism of the unclosing of the oocyst walls that is triggered by digestive agents, followed by the release of sporozoites from oocysts in the small intestine is still unclear [10,18]. Heather et al. first identified the profile of the proteome of sporulated oocyst walls and sporocysts/sporozoites, as well as an inventory of oocyst inner wall proteins and outer wall proteins [19]. Among the 1021 proteins characterized in sporocyst/sporozoite fractions and the 226 proteins detected in oocyst wall fractions, only 10 oocyst wall proteins were exclusively expressed in oocyst walls, and a family of tyrosine-rich proteins and late embryogenesis abundant domain-containing proteins likely played an acting role in conferring environmental resistance [19]. The asexual phase refers to endodyogeny, which means the parasite directly duplicates its monoploid genome, followed by the division of maternal cells into two daughter cells, not only leading to the production of tachyzoites in the acute infection of hosts but also causing the formation of cysts that contain many bradyzoites converted from tachyzoites in the persistent infection of hosts [1,20]. The life cycle of *T. gondii* was demonstrated to show a complex pattern of gene transcription and expression that was associated with each of its forms. Previous studies revealed that distinctive phase-specific mRNA, noticeable during the life history of *T. gondii*, reached 18% of the whole number of tags identified by SAGE [21]. Using high-throughput RNA-sequencing technology, Chen et al. compared the transcriptomes of the bradyzoites and tachyzoites of *T. gondii* and found that alternative splicing was differentially regulated during the life history of *T. gondii* [22]. The expression pattern of proteins between the three life history phases of *T. gondii* displayed significant differences, especially several virulent molecules and RP, which provided novel comprehension for the mechanisms of three life cycle phases adapted to their hosts [23].

Protein phosphorylation, a kind of PTM, was led by phosphatase acting on serine, threonine, tyrosine, and histidine on proteins. It not only was able to alter the activity, stabilization, degradation, and localization of a protein but also was a major switch mechanism for cell signaling process regulation by initiating or disrupting its interaction with other proteins within eukaryotic cells [24]. PTMs of proteins likely acted as key modulators of the biology, development conversions, and pathogenesis of apicomplexan parasites, and phosphorylation was no exception [25]. A phosphoproteomic analysis of the amastigotes and promastigotes of *Leishmania donovani* revealed that a great number of the phosphorylation residues (p-sites) were phase specific, and an “SF” motif, formerly characterized in trypanosomatid, was an important motif in the amastigotes of *Leishmania donovani* [26]. More than 30% of the predicted proteomes in *T. gondii* were covered by the highly dynamic and most prevalent PTM, phosphorylation, which was similar to *Plasmodium falciparum* [27,28]. A majority of the secreted proteins of *T. gondii*, especially ROPs and GRAs, were serine/threonine kinases and were indispensable for the virulence of *T. gondii*, for example ROP16, ROP18, and GRA24 [25,27,29,30,31,32,33,34,35]. Using iTRAQ-based phosphoproteomics technology, the phosphoproteome of the tachyzoites of *T. gondii* strains, falling into three classical genotypes, were compared with each other, and several motifs and protein kinases were overrepresented between *T. gondii* strains [36]. However, the differences of the phosphoproteomes between the different phases of *Toxoplasma gondii* remain a mystery.

The isobaric tag (IBT)-based quantitative phosphoproteomic approach has proved to be adequately accurate and sensitive to simultaneously identify and characterize the phosphoproteomes of several samples in an impartial way [37,38,39]. In our research, a quantitative phosphoproteomic approach, employing IBT integrated with TiO_2_ affinity chromatography, was applied to identify and analyze the differences of the phosphoproteomes between the sporulated oocysts and tachyzoites of the virulent ToxoDB#9 (PYS) strain of *T. gondii*. Our research first compares the phosphoproteome profiles of the developmental stages of *T. gondii* and lays a novel foundation for the elucidation of stage-specific differences in PTMs, which will possibly enhance the illustration of the regulatory style that is conducive to the development and infectivity of *T. gondii*.

## 2. Results

### 2.1. Phosphopeptide Profiling of Mature Oocysts and Tachyzoites of T. gondii

To gain a deeper understanding of the differences between the mature oocysts and tachyzoites of *T. gondii*, we used a quantitative phosphoproteomic approach based on IBT to compare the phosphoproteomes of mature oocysts and tachyzoites of the virulent *T. gondii* PYS strain. We obtained a total of 353,317 spectra from which 40,832 spectra were identified, at a false discovery rate (FDR) of 1%. Of the 10,645 phosphopeptides, the number of phosphopeptides harboring below 10 amino acids, 11 to 20 amino acids, 21 to 30 amino acids, 31 to 40 amino acids, and surpassing 40 amino acids were 1792, 7757, 1034, 50, and 12, respectively (Figure 1A). A total of 8181 nonredundant phosphorylation sites and 2792 phosphoproteins were characterized from the mature oocysts and tachyzoites of the virulent *T. gondii* PYS strain based on an FDR below 0.01 for phosphopeptides and a phosphoRS possibility above 0.75 for phosphosites. A large number of phosphopeptides harbored mono-phosphosites (10,206, 95.88%) and others had multi-phosphosites (439, 4.12%) (Figure 1B). Of the 8181 phosphorylation sites, 9574 (86.27%) were phosphorylated at serine (pSer), 1477 (9.28%) at threonine (pThr), and 47 (0.42%) at tyrosine (pTyr) (Figure 1C). The number of phosphoproteins including 1, 2, and at least 3 phosphosites were 1031, 550, and 1022, respectively (Figure 1D). Repetitiveness analysis denoted by the CV characterized 99% of phosphoproteins among the overall phosphorylated proteins when comparing S/T with the CV ≤ 0.4 (Figure 2).

### 2.2. Identification of Differentially Expressed Phosphopeptides (DEPs)

In total, 4059 phosphopeptides were determined to be DEPs, between mature oocysts and tachyzoites (|log1.5 fold change| > 1 and *p* < 0.05). Among them, 2597 phosphopeptides were upregulated and 1461 were downregulated (Figure 3A, Appendix A). The distribution of the quantities of DEPs in various degrees is shown in Figure 3B. Figure 4 displays the outcome of a hierarchical clustering analysis of DEPs between mature oocysts and tachyzoites.

### 2.3. Phosphorylation Motif Analysis for Quantitative Phosphopeptides

It was reported that motifs were indispensable for distinguishing zymolyte by the accordant kinases [40,41]. A motif analysis was used to explore the conservation of the cysteine array from −7 to +7 positions neighboring the phosphosites. Twenty-one motifs were extracted from the upregulated phosphopeptides in sporulated oocysts when comparing O/T, which contains 19 serine motifs (LxRxxS, GxxSP, KxxS, QxxSP, RSxSP, RxxS, SxxS, AxSP, ExSP, PxSP, DS, GS, SDxD, SDxE, SD, SExE, SGxE, SPxxxK, and SP), and two threonine motifs (GxxTP and TP) (Figure 5). The motif analysis of the downregulated phosphopeptides in sporulated oocysts when comparing O/T identified 13 serine motifs (ExxxxxxSP, KxxxxxxSP, LxRxxS, KxxS, PxxSP, RxxS, PxSP, ASP, GS, SPxxxT, SPxxxxR, SP, and SxxxxxK) and three threonine motifs (KxxT, RxxT, and TP) (Figure 6). Because each motif was consistent with one or multiple classes of kinases, the online software KinasePhos was applied to identify the classes of kinases corresponding to the motifs from the upregulated and downregulated phosphopeptides. The classes of kinases corresponding to the motifs from the upregulated and downregulated phosphopeptides are shown in Table 1 and Table 2, respectively. The categories of the kinases not only included traditional kinases, such as CK II, IKK, PKA, PKG, CDC2and, MAPK, but also some infrequent kinases, such as Ab1, EGFR, INSR, Jak, Src, and Syk. Thus, some differences of kinase secretory tendency and substrate-recognition preference of the sporulated oocysts and tachyzoites of *T. gondii* were indicated by those outcomes.

### 2.4. GO Category Analysis of DEPs

A GO analysis was conducted to survey the discrepancy of the function of DEPs between sporulated oocysts and tachyzoites. The upregulated and downregulated DEPs were enriched into three GO sorts, videlicet biological processes, cellular components, and molecular functions (Figure 7A,B). Metabolic processes, cellular processes, localization, biological regulation and the regulation of biological processes were the five most abundant GO terms under biological process for the upregulated DEPs. With regard to the cellular component category, cell parts, cells, organelles, membranes, and macromolecular complexes were the top five GO terms, including the most upregulated DEPs. The five most abundantly enriched GO terms for the downregulated DEPs involved cell parts, cells, organelles, macromolecular complexes, and membranes. Regarding the sort of the molecular function for upregulated and downregulated DEPs, catalytic activity, binding, transporter activity, structural molecule activity, and signal transducer activity were the five most prevalent GO terms.

Additionally, we investigated the relationship among GO terms using the co-expression network of GO terms associated with the three functional sorts produced by the RI graph package. As shown in Appendix A, 17 GO terms under the MF category, 3 terms (extracellular region, nucleolus, and myosin complex) under the CC category, and 39 terms under the BP category were significantly enriched in the GO network of upregulated phosphoproteins. However, 16 terms under MF, 16 terms under BP, and 6 terms under CC were remarkably upgraded in the GO network of downregulated phosphorylated proteins (Appendix A).

### 2.5. KEGG Pathway Analysis 

Among various organisms, enzymes harboring phosphorylation sites were capable of regulating central signal pathways [36]. The DEPs between sporulated oocysts and tachyzoites were plotted into pathways in the KEGG data to achieve a deeper understanding of the signal pathways regulated by phosphoproteins and to obtain intensive insights into the role of these DEPs. A total of 1223 DEPs were mapped to 143 pathways. Among them, RNA transport and phosphatidylinositol signaling (PI) systems were the two significantly enriched pathways that included 38 and 18 DEPs, respectively (Figure 8).

### 2.6. Interaction Network of Identified Phosphoproteins 

The protein–protein interaction (PPI) networks of interactors (combined score ≥ 0.9) between the DEPs were created by Cytoscape software to understand the mechanisms controlled by phosphorylation and to identify the associated operative bunches of the DEPs between the mature oocysts and tachyzoites of *T. gondii*. A total of 347 protein nodes and 724 interactor edges were included in the PPI networks (Figure 9). Some critical focuses were discovered in the PPI network, such as translation elongation factor 2 family protein (TGME49_205470), fibrillarin (TGME49_311430), nucleolar protein 5 (TGME49_205510), AARP2CN (NUC121) domain-containing protein (TGME49_313830), ribosomal protein RPL4 (TGME49_309120), and ribosomal protein RPL6 (TGME49_313390). Phosphoproteins were sorted into RNA splicing or translation, DNA replication or RNA transcription, mitochondria, histone, adapter, vacuolar, casein, nucleus, ribosome, kinases or enzymes, molecular chaperones, proteasome, or ubiquitin.

### 2.7. Kinase Connected Network

Because of the supposition that the behavior of the phosphorylated protein acting as a kinase was influenced by phosphorylation, a correlation analysis was conducted to characterize the phosphopeptides linked to the respective kinase. After removing the phosphorylated peptides with ≤~1.5-fold alteration profiles, we characterized 3 phosphorylated kinases associated with 323 phosphopeptides (Appendix A). Positive and negative correlations denoted that the kinase beneficially or negatively impacts its action, causing downstream influence, just as the kinase phosphorylation states through phosphorylation. The underlying disparity of the kinases and relevant substrates between mature oocysts and tachyzoites of the *T. gondii* strain epidemic in China was suggested by these correlations. Among the two clusters of phosphopeptides, the peptides associated with *T. gondii* S8GFS8 (DNMT1-RFD domain-containing protein) and *T. gondii* S8F5G5 (Histone kinase SNF1) occupied the hub combining the most kinase peptides (280 phosphopeptides and 42 phosphopeptides, respectively) (Figure 10). One of the most connected kinases peptides (Histone kinase SNF1) was annotated in the molecular function GO terms associated with binding, catalytic activity, protein kinase activity, kinase activity, transferase activity, calmodulin-dependent protein kinase activity, protein serine/threonine kinase activity, and transferring phosphorus-containing groups.

### 2.8. Potential Phosphor-Dependent Interactions

To characterize the assumed role of the different phosphosites in our phosphoproteomic profile and forecast the influence of phosphorylation or mutation on the interplay between 2 phosphoproteins (activated or inactivated), 31 interactions were discovered by Mechismo (Appendix A) and phosphorylation sites of 6 *T. gondii* proteins were deemed as putative enablers or disablers of protein interactions after being plotted backward to *T. gondii* proteins (Figure 10). Nine interactions and twelve interactions were affected by the S7 phosphosites of *T. gondii* A0A125YH18 gene and S7 phosphosites of *T. gondii* A0A125YY77 gene, respectively. The S9 phosphosites of the *T. gondii* A0A125YKL3 gene impacted three interactions, and the S10 phosphosites of the *T. gondii* A0A125YR63 gene and S1 phosphosites of the *T. gondii* S8F1R4 gene enabled one interaction and disabled one interaction. The S3 phosphosites of *T. gondii* A0A125YNJ9, S10 phosphosites of *T. gondii* A0A125YNH2, and T3 phosphosites of *T. gondii* A0A125YXW1 simultaneously affected one interaction.

## 3. Discussion

PTMs on proteins transmit signals that are exchanged between parasites and their hosts, and they play a key role during the process of parasitic development [26,42]. Genotype Chinese 1 (ToxoDB#9) *T. gondii* is the preponderant genotype in Asia and is dominantly prevalent in China. It accounts for almost 78% of *T. gondii* strains in China and has a semblable virulence to type 1 RH [7,43,44,45]. To investigate the difference of phosphorylation on proteins, a key PTM that regulates apicomplexan biology [25], across the life cycle of genotype Chinese 1 (ToxoDB#9) *T. gondii,* an iBT-based LC-MS/MS connected with TiO_2_ chromatography was applied to study the phosphoproteomic plot of mature oocysts and tachyzoites of the virulent genotype Chinese 1 (ToxoDB#9) *T. gondii*. Altogether, 10,645 unique phosphopeptides, 8181 nonredundant phosphorylation sites, and 2792 phosphoproteins were characterized in the present research. A total of 4059 phosphopeptides were identified as DEPs when comparing oocysts to tachyzoites, including 2598 upregulated phosphopeptides and 1461 downregulated phosphopeptides. Although progress in phosphoproteomic approaches has gradually enhanced the phosphoproteomic study of the developmental phases of *T. gondii*, the illustration of the major effects of phosphorylation for *T. gondii* development remains limited. The quantitative phosphoproteomic analysis between the tachyzoite and early bradyzoite identified 7650 phosphosites on 2235 *T. gondii* proteins, and 144 phosphorylation sites and 193 phosphorylation sites on 170 proteins were deemed to be differentially altered in the tachyzoite and bradyzoite, respectively. Out of these, 51 phosphorylated proteins were predictively secretory proteins [46]. He et al. compared the phosphoproteome of initially penetrating tachyzoites and pre-egressed tachyzoites, employing an iTRAQ-based quantitative approach. Altogether, 42 phosphosites on 38 *T. gondii* proteins were characterized, of which 10 phosphoproteins were deemed to be significantly differential proteins in their phosphorylation level [47]. Our results identified more phosphoproteins in tachyzoites than in previous studies. It is not unexpected that several phosphoproteins known to regulate processes critical for *T. gondii* development were characterized in our study, such as CDPK7 and αSNAP. Not only was the event of TgCDPK7 phospholating TgRab11a critical for *T. gondii*’s development and parasitic protein trafficking, but so were TgCDPK7-regulated phosphatidylethanolamine (PE) levels in *T. gondii* [48]. Similarly, not only was the phosphorylation of αSNAP necessary for secretory organelle biogenesis and the formation of organelles in *T. gondii*, but αSNAP controlled the disassembly of *cis*-SNARE complexes through the phosphorylation of *cis*-SNARE [49]. A comparative phosphoproteomic analysis of the tachyzoites, cysts, and oocysts of *T. gondii* should be more useful in the understanding of the developmental biology and stage-specific molecular determinants, in light of the ability of *T. gondii* to differentiate from tachyzoites to bradyzoite-containing cysts as a key role in their pathogenesis [24,50]. Because the virulent strain of *T. gondii* was generally deemed to have lost its ability to form a cyst [48,51], a pure cyst of the virulent ToxoDB#9 (PYS) strain of *T. gondii* was not successfully obtained either in mice or in guinea pigs, and the cyst was not included in the life-cycle stages of the virulent ToxoDB#9 (PYS) strain of *T. gondii* in our study. Although it was reported that the cyst of a virulent RH strain of *T. gondii* might be produced in mice treated with SDZ, the cyst of type one strains cannot be obtained in non-immune and non-treated mice, which means that conducting research associated with the cyst of a virulent strain of *T. gondii* during the natural course of infection will still be unavailable [48,52]. The sporozoites excysted from the oocyst and sporocyst walls, rather than sporulated oocyst, invaded the enteroepithelial cells of the intestine of the host, followed by the differentiation of tachyzoites [10]. It is noteworthy that the critical structure of the oocyst wall provides protection to the sporozoites until its release, and research involving the structure and molecules of the oocyst wall could support the detection and elimination of *T. gondii* oocysts in various environmental matrices [10,16,19]. Therefore, research on the structure and molecules of oocyst walls is urgently needed, especially PTMs on OWP (oocyst wall protein) because of their important function on protein, such as signal transduction (phosphorylation), half-life alteration (glycosylation), and membrane anchoring (acetylation) [53,54]. Additionally, it is inevitable that pure sporozoite fractions were contaminated with oocyst wall fractions, due to sporozoite fractions being deposited in oocyst walls in a continuous manner, according to a previous study on the proteome of the sporozoite and oocyst wall of *T. gondii* [19]. So, it is likely that the sporozoite cannot really be separated from oocyst walls. In our data, an OWP (TGME49_268225) was found to be phosphorylated on two serines, which verified the presence of phosphorylation in OWP. This had only been predicted in previous research [53]. The role of this phosphorylated OWP in the function of the oocyst wall is still a mystery.

### 3.1. Motifs

The seven motifs that were synchronously identified from the upregulated phosphopeptides and downregulated phosphopeptides when comparing O/T, and SP and RxxS were two basic motifs. The SP motif recognized by the MAPK and the RxxS motif was the recognition site of Ca^2+^/Cam-dependent protein kinase and 14-3-3 proteins [40,55,56]. Fourteen motifs (GxxSP, QxxSP, RSxSP, SxxS, AxSP, ExSP, DS, SDxD, SDxE, SD, SExE, SGxE, SPxxxK, and GxxTP) were unique in upregulated phosphopeptides and nine motifs (ExxxxxxSP, KxxxxxxSP, PxxSP, ASP, SPxxxT, SPxxxxR, SxxxxxK, KxxT, and RxxT) were exclusive to downregulated phosphopeptides. These results indicated the discrepancy of consensus sequences and phosphatases that were respectively activated and inhibited during *T. gondii* development. 

The classes of kinases corresponding to motifs from the upregulated and downregulated phosphopeptides identified using KinasePhos not only contained traditional kinases but also included some infrequent kinases. In *T. gondii*, PKA and PKG were the subsequent products of cyclic nucleotide signaling that had inducing and inhibitory effects on bradyzoite differentiation, and suppression of PKA and PKG brought on the expression of bradyzoite genes and laggard propagation [57]. Further, PKA acted as a suppressor of PKG-regulated signaling to form an interplay between cAMP and cGMP to govern timely pH-dependent egression of *T. gondii* [58]. Although the role of PKC in *T. gondii* remains unreported, it is known that PKC possibly plays a part in larval growth and regulates mating and egg discharge of adults in some worms, such as *schistosome* [59]. MAPKs of *T. gondii* included three members named signal-regulated 7 (ERK7), MAPKL1, and MAPK2. ERK7 regulated conoid biogenesis and MAPKL1 mediated centrosome replication. Absent localization of MAPK2 at the centrosome was observed during the cell cycle of MAPK2-deficient *T. gondii*, which indicated that the commencement of mitosis and fulfillment of the centrosome in parasites was requested for the process regulated by MAPK2 [60]. CKI was a long-sought-after protein kinase thought to impact cell cycle progression by means of facilitating the destruction of CDC 25A [61]. As for infrequent kinases, EGFR and INFR were two kinds of conserved kinases and potentially indicated function in the regulation of development and host–pathogen molecular interplay in parasites [62]. Jak1 and Jak2 were components of the INF-γ signaling cascade that was essentially required by the anti-Toxoplasma activity inside human cells [63]. Src tyrosine kinase phosphorylated the central SD (substrate domain) of P130Cas (Crk-associated substrate) to control cell migration, which generated a coordinated cellular response [64]. Ab1 had the ability to bind to chromatin to interact with Atm and Atr in response to DNA damage. However, the role of Ab1 in cell cycle regulation and DNA restoration continues to be controversial [65].

### 3.2. GO and KEGG Analysis

The PI signaling system and RNA transport were two significantly enriched pathways. Among the 18 phosphoproteins annotated to the PI signaling system pathway, 7 phosphoproteins (TGME49_238400, TGME49_230490, TGME49_245730, TGME49_296010, TGME49_248830, TGME49_316230, and TGME49_215700) were reported to be phosphoinositide kinases and phosphatases in *T. gondii*, which were PI monophosphates, and whose annotations included a PI3-kinase, a PI4-kinase class IIIβ, two PI4-phosphate 5-kinases, a PI-specific phospholipase C, an inositol-polyphosphate 5-phosphatase, and an inositol-phosphate phosphatase [66]. Although the expression of PI3-kinase was low in the tachyzoite of *T. gondii*, depletion of this kinase resulted in apicoplast loss and death of the tachyzoite [66,67]. Additionally, PI3-kinase bound to *Tg*PROP1 and *Tg*PROP2, two proteins belonging to PROPPIN homologs, and had lipid-binding properties that were vital for their correct membrane targeting [68]. These studies demonstrated that PI3-kinase was crucial in *T. gondii* replication and apicoplast biogenesis. PI4-phosphate 5-kinase had two isoforms, and one isoform (TGME49_230490) identified in our study was predicted to contain an N-terminal calcium sensor domain named EF-hand [66]. Interestingly, this isoform was also reported to be essential during the infection of human fibroblasts by *T. gondii* [66,69]. A PI synthase (PtdIns synthase, TGME49_207710) identified in the present study was found to be annotated to the pathway PI signaling system. *T. gondii* PtdIns synthase possessed a catalytically vital CDP-diacylglycerol phosphotransferase motif in the Golgi apparatus and knockout of PtdIns synthase led to abrogation of the lytic cycle of *T. gondii* in human fibroblasts [70]. Furthermore, *T. gondii* PtdIns synthase participated in the biosynthesis of PtdIns-anchored proteins, which were important for the adhesion and invasion of *T. gondii* tachyzoites within the parasitic cells [71,72]. These two aspects of PtdIns-anchored proteins during the life cycle of *T. gondii* indicated that *T. gondii* PtdIns synthase plays a key role during the development of *T. gondii*.

### 3.3. PPI Network

Besides some familiar functional clusters harboring phosphorylated proteins, such as RNA splicing or translation, DNA replication, or RNA transcription, ribosome, kinases, or enzymes, some novel bunches were characterized in the PPI network, especially the histone, casein, proteasome or ubiquitin, adapter, and molecular chaperones. Like other eukaryotic parasites, histones of *T. gondii* were also the target of plentiful PTMs, and phosphorylation was conserved during the life-cycle phases of *T. gondii*. Nonetheless, dynamic changes of histone PTM regulated the chromatin structure and gene expression between developmental phases of *T. gondii* [73]. Quantitative changes in histone phosphorylation between tachyzoite and sporulate oocysts were included in our phosphoproteomic data. It seems that casein kinase (CK) has a biological function during *T. gondii* infection or development. TgCK1 included two isoforms, TgCK1α, showing enzyme activity, and TgCK1β, without enzyme activity. TgCK1α positively regulates the expression of TgROPs, and the deletion of TgCK1α enhances acute virulence in mice via inhibition of IL-12 generation, activation of STAT3, and the incremental expression of ROPs. It remains a mystery whether TgCK1 has functions between the developmental phases of *T. gondii* [74,75]. TgCK2 was able to influence the PI3K-AKT-mTOR pathway responsible for sensing the environment inside host cells and signaled the parasites to replicate or to transform into bradyzoites [76]. The proteasome was present in various protozoan parasites, as well as in *T. gondii*. The *T. gondii* proteasome was proved to degrade complex cellular structures and to be essential for its in vitro replication [77,78]. It was proved that the ubiquitin proteome of *T. gondii* tachyzoites contained an excess of 500 ubiquitin-modified proteins and nearly 1000 ubiquitin-modified sites, and 35% of these ubiquitin-modified proteins were cell cycle-mediated, which interplayed with phosphorylation in the final S/M phase to regulate the cell cycle of *T. gondii* [79]. The adaptor compound is responsible for the reconstruction of the mitochondria membrane and replication of the mitochondrion greatly affects the endodyogeny of tachyzoites of *T. gondii* [80]. Like other protozoan parasites, *T. gondii*’s molecular chaperones also exhibited a crucial role in the growth and pathogenesis of the parasite [81,82]. Hsp60 localized to the mitochondrion and played a function in growth [82,83]. Hsp70 is a crucial molecule for the intracellular multiplication of *Toxoplasma gondii*, and Hsp90 was associated with several stages of the parasite, for example, host-cell penetration, multiplication, and tachyzoite–bradyzoite inter-transformation [84,85]. Intriguingly, molecular chaperone formed chaperone networks and chaperone-co-chaperone partnerships to mediate the activity of client chaperones for the purpose of having a role in the differentiation of *T. gondii*, for example Hsp70 and Hsp90 [82].

### 3.4. Kinase–Protein Interactions

Among the latent kinase–protein interactions forecasted in our research, the histone kinase SNF1 (TgH3 and S8F5G5) was one of the greatest connected peptides. The phosphorylated site in threonine was characterized in TgH3 in a previous study [73], which was in line with our study. However, the detailed function of phosphorylated modification in threonine remains unknown [73]. Three conserved phosphorylated modifications in TgH3 were deemed to have a function in mitosis and interphase of *T. gondii* [73,86]. Additionally, TgH3 worked in concert with TgGCN5, based on a modification to mediate gene transcription and expression relevant to the division of *T. gondii*, which was a well-known function in *Saccharomyces cerevisiae* [87,88,89]. Actin ACT1(A0A125YH18), another greater connected peptide, was demonstrated to be indispensable for the effective invasion of *T. gondii* into host cells [90].

## 4. Material and Methods

### 4.1. Ethics Statements

Each animal experiment was scrutinized and consented to by the Animal Administration and Ethics Committee of Gansu Agricultural University and was strictly conducted based upon the proposal expounded in the Animal Ethics Procedures and Guidelines of the People’s Republic of China. All efforts were made for the diminution of the suffering of animals and the reduction of the number of animals used during the experiment.

### 4.2. Mice, Guinea Pigs, Cats, and Parasite Strains

SPF female BALB/c mice (6 to 8 weeks old), SPF female guinea pigs (16 weeks old), and SPF female kittens (10 weeks old) were supplied by Laboratory Animal Center of LVRI. The tachyzoites of the Chinese 1 (PYS, ToxoDB#9) strain of *T. gondii* were passaged in mice for preservation in our laboratory.

### 4.3. Preparation of Tachyzoites and Sporulated Oocysts

#### 4.3.1. Collection and Purification of Tachyzoites

Tachyzoites of *Chinese 1 T. gondii* were unfrozen from liquid nitrogen and then were passaged in mice for three generations to regain viability. Afterward, SPF BALB/c mice inoculated with ~100 tachyzoites were continually raised in the Laboratory Animal Center. Infected mice were sacrificed when they displayed significant symptoms of *T. gondii* infection, for example ruffled fur, reduction of appetite, and head slouching. The tachyzoites collected from peritoneal washing and rinsing of the peritoneal cavity of mice with sterile PBS were repeatedly isolated and centrifugated at 1680× *g* for 15 min for 3 washes with PBS. After the supernatant was discarded, the final sediment containing tachyzoites was purified by digestion of the host cells at 37 °C for 20 min using 0.25% trypsin. After centrifugation at 1680× *g* for 15 min, the epipelagic liquid was thrown away and the parasite sediment was resuspended using 1 mL PBS in a tube and stored at −80 °C.

#### 4.3.2. Collection and Purification of Sporulated Oocysts

Collection, purification, and maturation of unsporulated oocysts of the PYS strain were conducted, as previously described [23]. In summary, about 10^5^ tachyzoites of the PYS strain were used to infect 13 week old SPF female guinea pigs through intraperitoneal injection, and cysts in the muscle and brain tissues of euthanized guinea pigs were fed into specific-pathogen-free kittens after guinea pigs one month later. The oocysts shedding from the kittens were gathered every day to purify unsporulated oocysts by the density gradient centrifugation method using CsCl [23,91]. Then, the purified oocysts were pelleted and blended with 2% H_2_SO_4_ to sporulate in an aerobic condition. Finally, the sporulated oocysts were rinsed with 0.85% saline and then preserved at 4 °C until use.

### 4.4. Extraction and Digestion of Protein

Triple samples from mature oocysts and tachyzoites of the *T. gondii* PYS strain were employed to extract total protein. The protein-separating solution (20 mM Tris-HCl, 7 M urea/2 M thiourea, 10 mM DTT, and 0.2% SDS, pH 8.5) containing PhosSTOP (phosphatase inhibitors, Roche, Basel, Switzerland) and PMSF (1 mM, Thermo Scientific, Chelmsford, MA, USA) was blended into mature oocysts (~10^7^) and tachyzoites (~10^8^) for the lysis of parasites. Afterward, samples were fractured using a sonication assay (2 s ON and 3 s OFF cycle, 5mins) on ice and then were pelleted at 25,000× *g* for 20 min at 4 °C to eliminate lysate scrap. DTT (10 mM, 60 min, 56 °C) and IAM (55 mM, 45 min, 37 °C) were applied to deoxidize and alkylate samples, respectively. Subsequently, a BCA assay kit (QuantiProTM, Sigma-Aldrich Co. LLC, St. Louis, MO, USA) was used to determine the concentration of protein. To cleave protein, trypsin (Promega, Madison, WI, USA) was added into 300 μg of protein of each repeat with a protein-to-enzyme ratio of 40:1, and the mixture was retained at 37 °C for 12 h. The digestion was terminated by FA at a concentration of 0.5% (*v*/*v*), and Strata X solid-phase extraction pillars (Phenomenex, Torrance, CA, USA) were then used to desalt the samples. Last, a SpeedVac was used to dry the samples, and the samples were applied for IBT labeling.

### 4.5. IBT Labeling

A 6-plexed IBT agent (BGI-Shenzhen, Shenzhen, China) was applied to label peptides. Briefly, a labeling reagent mixed with 50 µL of isopropyl alcohol was mingled with vacuum-dried peptides, and then the blend was incubated at ambient temperature for 2 h. After that, the labeled peptides were desalinized and condensed using Sep-Pack C18 Cartridges (Waters, Worcester, MA, USA). Peptides from sporulated oocysts were labeled using 114, 115N, and 115C, and peptides from tachyzoites were labeled with 117C, 118N, and 118C. 

### 4.6. Enrichment of Phosphopeptides by Affinity Chromatography

The phosphopeptides of labeled peptides were enriched using TiO_2_ (GL Sciences, Tokyo, Japan) pellets, as depicted previously [36]. First, a suspension of TiO_2_ beads and 1 mL of loading solution (2% glutamic acid, 65% ACN, and 2% TFA) was used to mix IBT-labeled peptides at a beads-to-peptides ratio of 4:1(mass/mass). Then, the blend was shaken by a gyrator at 37 °C for 1 h, and centrifugation (12,000× *g* for 5 min) was applied to precipitate the mixture. After discarding the supernatant, the precipitation was cleansed using 500 μL of elution solution 3 times, as described above. Subsequently, a 600 μL elution buffer was applied to suspend the sediment, followed by pendulation for 20 min. Post centrifugation at 12,000× *g* for 1 min, the sediment was washed with a 500 μL elution solution, as described above, and was used for the LC-MS/MS analysis. Finally, vacuum-dried phosphopeptides were gathered and provided for analysis by LC-MS/MS.

### 4.7. LC-MS/MS

A Q Exactive Mass Spectrometer (Thermal Scientific, Chelmsford, State of Massachusetts, USA), coupled with HPLC (Phenomenex columns Gemini-NX 3u C18 110A 150 × 2.00 mm), was used to conduct an LC-MS/MS analysis of the IBT-labeled peptides. First, the IBT-labeled peptides were fractionated by a High pH Reversed-Phase Peptide Fractionation Kit (Thermo Scientific Pierce, Chelmsford, MA, USA), in accordance with the producer’s directions. In summary, phosphopeptides were unfrozen using 300 μL of 0.1% TFA and were laden into a forehand-balanced fractionation pin pillar. Following the phosphopeptides being combined with the resin, desalting of the peptides was performed using a wash by sterile water centrifugated at a low speed. Elution of the combined peptides was conducted by a volatilizable high-pH elution that involved an increasing concentration gradient of ACN. The eluted phosphopeptides were gathered into six proportions and were vacuum-dried (Thermo Scientific, Chelmsford, MA, USA) using centrifugation. Afterward, peptides solubilized by 20 μL of solution A (2% ACN and 0.1% FA) were uploaded into a C_18_ trap pillar and the velocity of loading was set at 8 μL/min for 4 min. The elution of the samples was performed by a sequential concentration gradient of solution B (95% ACN and 0.1% FA) at a speed of 300 NL/min. The concentration of solution B was initially 5% and ascended to 35%. Then, the concentration of solution B progressively ascended into 60% in 5 min and rose to 80% within 2 min. After the preservation of solution B at 80% for 2 min, the concentration of solution B was decreased to 5% in 1 min, followed by a maintenance of 5% solution B for 10 min. Finally, the spectra of the original-rank MS were acquired with the listed directions: scan scope = 350 to 1500 *m*/*z*, resolution = 70,000, maximum injection time = 40 ms, NS AGC target = 3e6. The parameters of the spectra of the second-grade MS were set as follows: resolution = 17,500, maximum injection time = 40 ms, and AGC target = 1e5. The pattern of high-energy collisional dissociation was applied to screen out the 20 top precursors. 

### 4.8. Data Analysis

The databank (http://www.toxodb.org/common/downloads/release-10.0/ Toxoplasma gondii ME49/fasta/data/, 8 March 2019) search was carried out using Proteome Discoverer 1.4 (Thermo Scientific, Chelmsford, MA, USA) integrated with a Mascot retrieval tool (version 2.3, Matrix Science). The following patterns were included in the search: enzyme = trypsin; peptide mass tolerance = 20 ppm; fragment mass tolerance = 0.05 Da; fixed modification = IBT-10plex (N-term), carbamidomethyl (C), and IBT-8plex (K); variable modification = oxidation (M), acetyl (protein N-term), and deamidated (NQ); phosphorylation (S/T/Y) maximum missed cleavage = 2. The identification and quantification of phosphopeptides were conducted on the basis of *p* ≤ 0.05, and the reliability of phosphosites was based on a phosphoRS probability of not less than 0.75. Phosphopeptides with a fold-change > 1.5 or <0.75 were deemed as DEPs. The clustering of DEPs was performed by a hierarchical clustering assay, and maps of the cluster were produced using a software package called MeV (Multiple Experiment Viewer, version 4.9.0).

### 4.9. Bioinformatics

The prediction of phosphorylation motifs of phosphopeptides was conducted to search for conservative motifs approximal to phosphosites using the network software Motif-X (http://motif-x.med.harvard.edu/, 10 March 2019). Briefly, the phosphopeptides identified in our study, harboring the extent of ±7 aa from the central phosphosites location beyond vague phosphosites, were input into the Motif-X software, and conservative motifs approximating phosphosites were extracted. The motifs extracted in our research were displayed with logos and represented by graphics. A GO analysis (http://www.geneontology.org, 25 August 2019) was conducted to categorized the DEPs into three functional sorts (cell component, biological process, and molecular function). A pathway annotation of the DEPs was performed by a KEGG analysis (http://www.genome.jp/kegg/, 10 March 2019), and bubble charts were applied to manifest those significantly enriched KEGG pathways. The STRING (http://string-db.org/, 10 March 2019) database was exerted to search the interaction networks of the DEPs to investigate the biological processes mediated by phosphorylation and an analysis of the interactive networks of the DEPs contributed. Last, interactive partners included in the PPI networks whose combined scores were not less than 0.7 were presented by Cytoscape (version 3.0.2).

### 4.10. Relevance Analysis of T. gondii Protein Kinase with Phosphorylated Peptides

Using the Pfam 31.0 database combined with HMMER (http://hmmer.org, 16 January 2021) [92], kinases within Pfam 31.0 Informix associated with phosphorylated proteins in the present study were identified. The content of phosphopeptides associated with protein kinase and protein tyrosine kinase domains was acquired and the correlation between the kinase-associated phosphorylated peptides and the phosphorylated peptides of our dataset was performed using a Pearson correlational analysis. Partners of the kinase-associated phosphorylated peptides and phosphorylated peptides with correlations >0.998 were picked up to create a network by Cytoscape 3.6.0 (https://cytoscape.org, 6 August 2021).

### 4.11. Forecast of Protein Coactions Activated or Inactivated by Phosphorylation

The software Mechismo was applied to search the identified phospho-activated and phospho-inactivated protein interactions for the purpose of obtaining the interactions activated or inactivated by phosphorylation [93]. Because of the absence of *T. gondii* proteomes within Mechismo, the eggNOG-mapper (http://eggnogdb.embl.de/#/app/ emapper, 29 October 2019) was employed to obtain the human and yeast homologs of the phosphoproteins in our research. The associated OG were initially acquired, and BioPython software (http://biopython.org/, 29 October 2019) was used to seek out their yeast and human orthologs. Afterward, identical phosphosites within the human or yeast proteins were extracted by aligning all the homolog arrays to the accordant *T. gondii* protein by applying the Clustal Omega software [94]. Additionally, the extracted proteins were inputted into Mechismo (http://mechismo.russelllab.org/, 16 January 2021) to investigate the homologs of the protein interactions that were activated or inactivated by phosphorylation in the human and yeast interconnection. Last, *T. gondii* proteins matched with orthologues were screened out.

## 5. Conclusions

Our research first uncovered the global phosphoproteomic differences between tachyzoites and sporulated oocysts of the *T. gondii* epidemic in China using the IBT technology integrated with TiO_2_ affinity chromatography. The obvious discrepancy in the conservative motifs, functional GO terms, enriched pathways, and the protein–protein network was indicated in the phosphopeptides between the tachyzoites and sporulated oocysts of *T. gondii*. The kinase-connected network analysis revealed the function of phosphorylation across the life cycle of *T. gondii*. An unknown protein (S8GFS8) and histone kinase SNF1 (S8F5G5) occupied the position of peptides connecting the most kinases, which played a key role in the differentiation of *T. gondii*. Our results will promote a better comprehension of the phenotypic differences between the tachyzoites and sporulated oocysts of *T. gondii*.

## Figures and Tables

**Figure 1 molecules-27-01022-f001:**
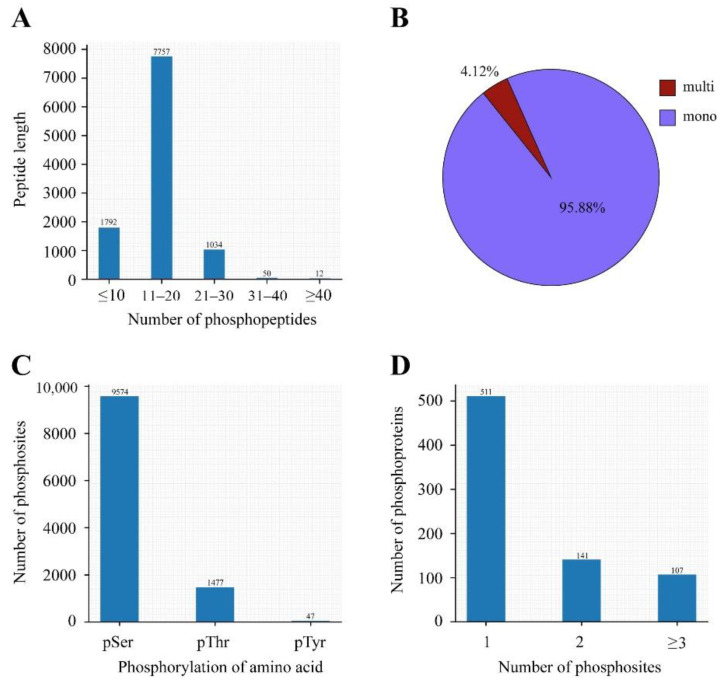
Large-scale information regarding the phosphoproteomic research between sporulated oocysts and tachyzoites of the *T. gondii* PYS strain. (**A**) The distribution of phosphopeptides on different peptide lengths. (**B**) The proportion of mono-phosphosites and multi-phosphosites among total phosphopeptides showed that a large part of the phosphopeptides have single phosphosites. (**C**) Proportion of the pSer/pThr/pTyr phosphoproteome. Phospho-pSer, phospho-pThr, and phospho-pTyr accounted for 86.27% of all phosphosites, 9.28% of all phosphosites, and 0.42% of all phosphosites, respectively. (**D**) Distribution of phosphosites on phosphoproteins. Most phosphoproteins included single phosphosites.

**Figure 2 molecules-27-01022-f002:**
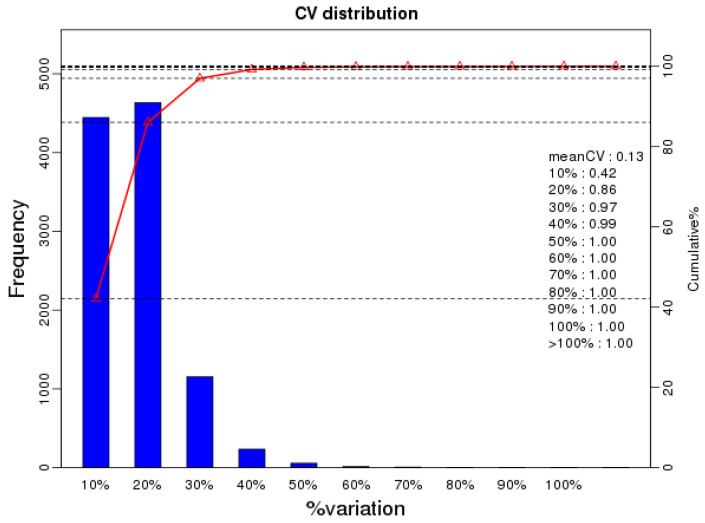
Repeatability analysis of the phosphopeptide quantification founded on the value of CV (coefficient of variation). The *x*-axis represents the percentage values of CV. The quantities of phosphopeptides and the accumulative proportion of phosphopeptides are indicated by the left and right *y*-axis, respectively.

**Figure 3 molecules-27-01022-f003:**
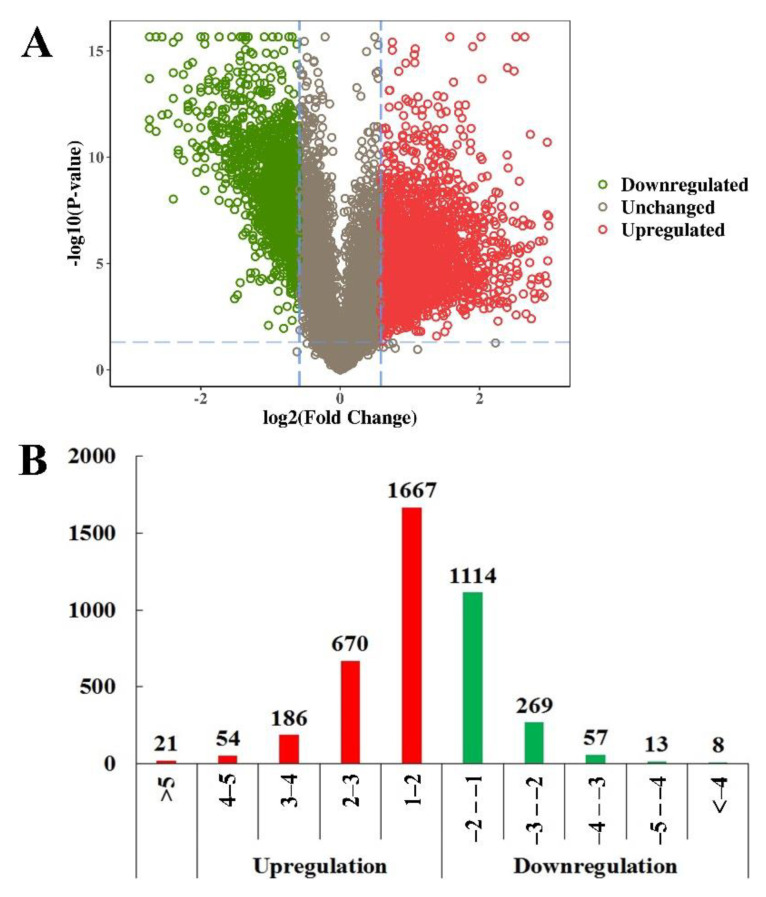
Quantitative analysis of characterized phosphopeptides. (**A**) Scatter plot of quantified phosphopeptides. The log_2_ fold change value and −log_10_ *p* value of phosphopeptides are indicated by the *x*-axis and the *y*-axis, respectively. The red circles and green circles denote upregulated and downregulated DEPs, respectively. (**B**) Distribution of upregulated DEPs and downregulated DEPs between the sporulated oocysts and tachyzoites on different levels. The *x*-axis and *y*-axis indicate the values of changes and the quantity of DEPs. The red color and green color represent upregulated DEPs and downregulated DEPs, respectively.

**Figure 4 molecules-27-01022-f004:**
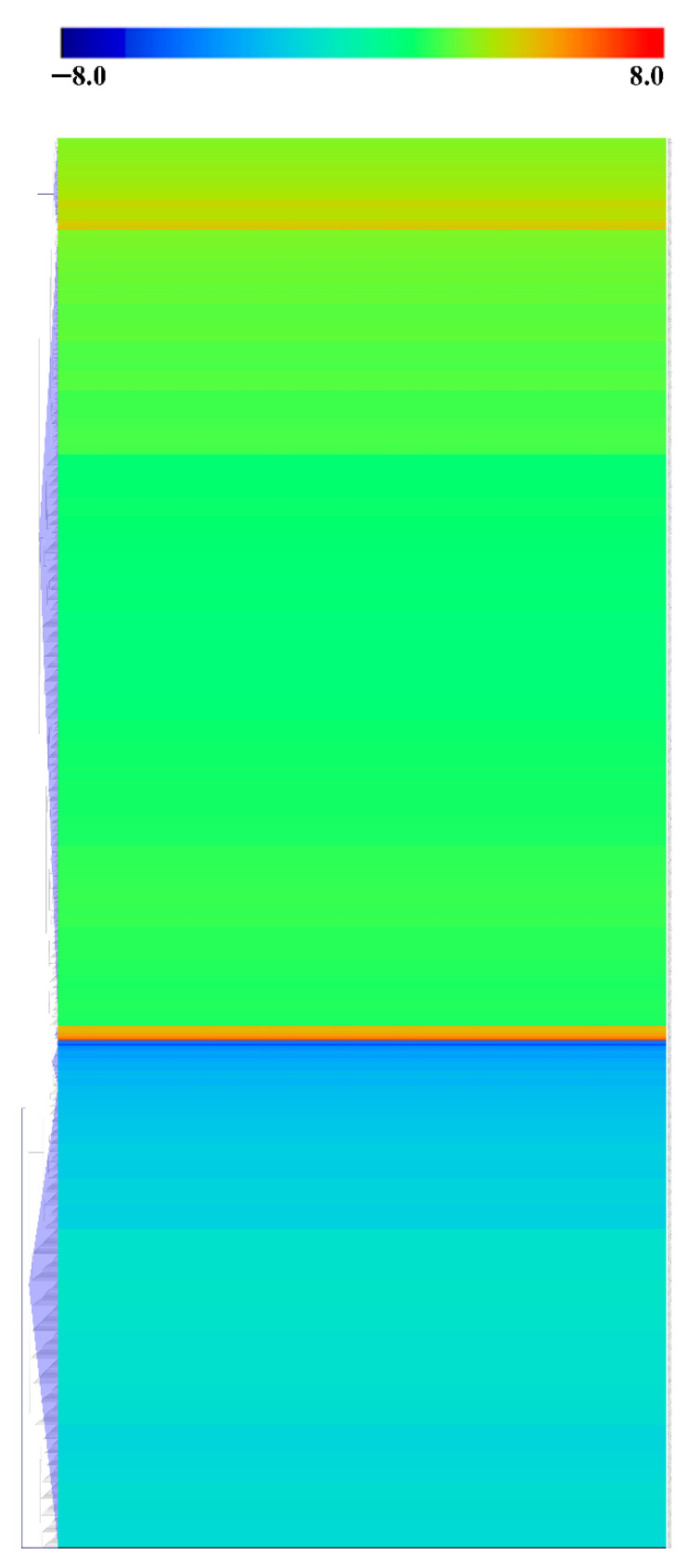
Hierarchical clustering analysis of DEPs. Log1.5 fold change value of DEPs is denoted by different colors.

**Figure 5 molecules-27-01022-f005:**
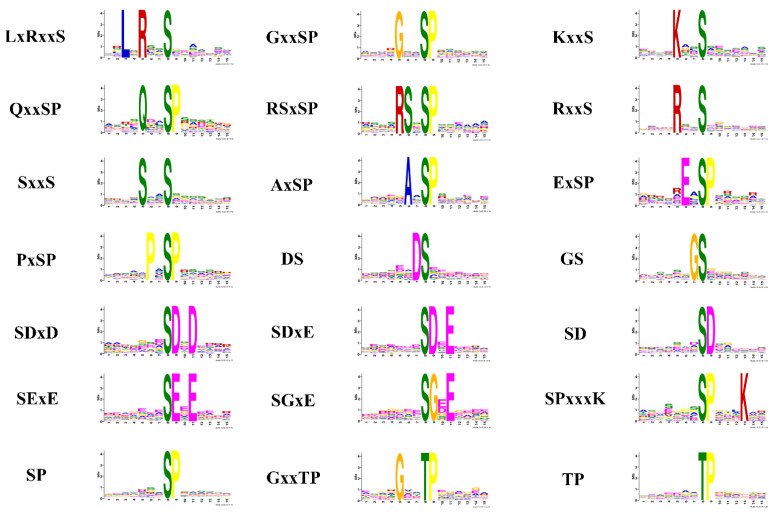
Phosphorylation site motif analysis of the upregulated DEPs using Motif-X. The degree of amino acids emerging at specific locations is represented by the elevation of the symbols.

**Figure 6 molecules-27-01022-f006:**
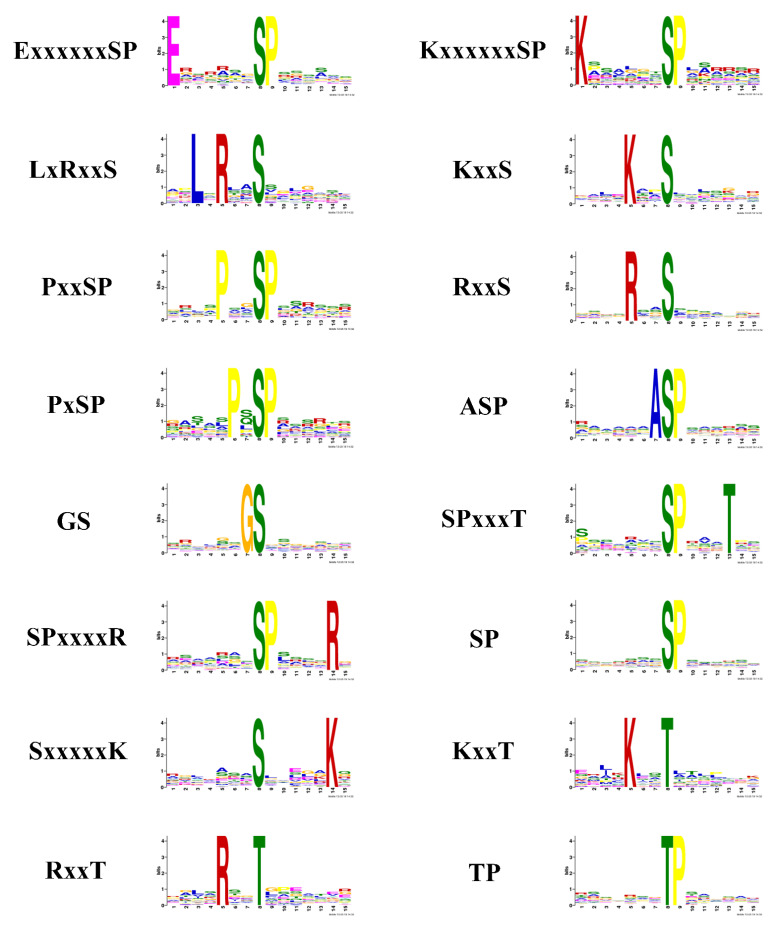
Phosphorylation site motif analysis of the downregulated DEPs using Motif-X. The degree of amino acids emerging at specific locations is represented by the elevation of the symbols.

**Figure 7 molecules-27-01022-f007:**
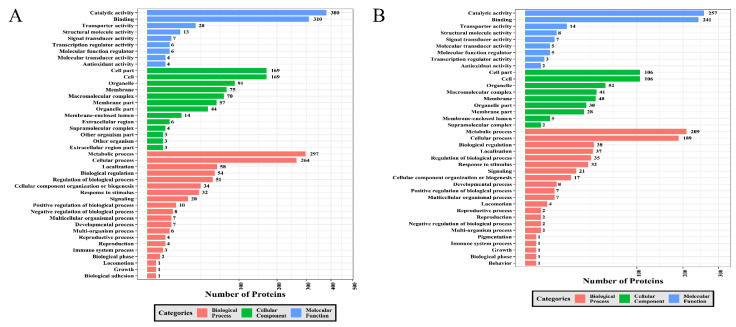
GO enrichment analysis of DEPs. (**A**) GO enrichment analysis of upregulated DEPs. (**B**) GO enrichment analysis of downregulated DEPs. The number of DEPs and GO terms are displayed on the horizontal axis and vertical axis, respectively.

**Figure 8 molecules-27-01022-f008:**
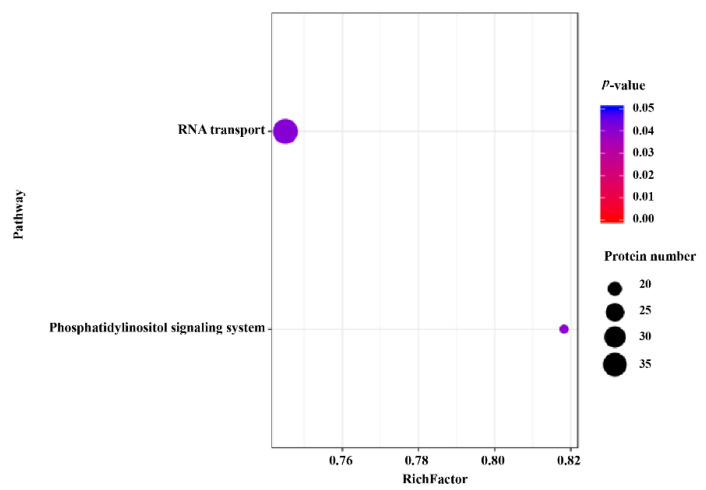
KEGG pathway enrichment analysis of DEPs between sporulated oocysts and tachyzoites. The *y*-axis indicates the KEGG pathways DEPs significantly enriched and the *x*-axis denotes the rich factors of the pathways. Rich factor implies the proportion of DEPs among total phosphoproteins in the pathway. Greater degrees of enrichment are indicated by higher rich factors. The color and size of the node represent the *p* value and quantity of phosphoproteins of the corresponding pathways.

**Figure 9 molecules-27-01022-f009:**
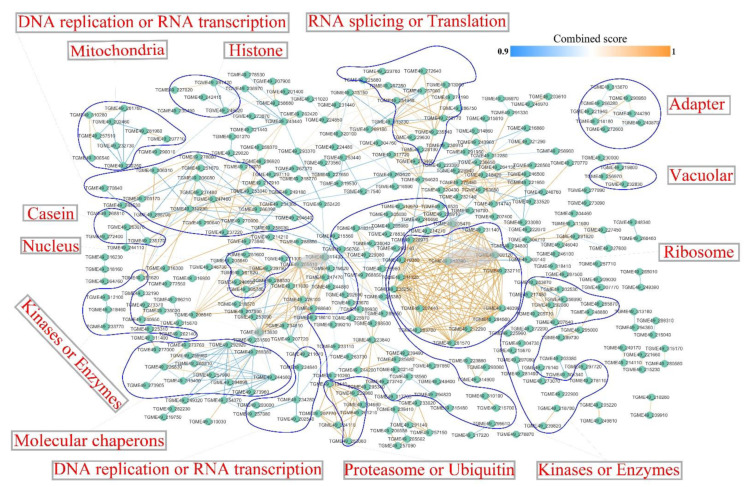
Protein–protein interaction (PPI, combined score ≥ 0.9) network of the DEPs between sporulated oocysts and tachyzoites. The nodes indicate DEPs and the lines between the nodes refer to the interactors between DEPs. The color of the edge symbolizes the combined score of the interacting partners.

**Figure 10 molecules-27-01022-f010:**
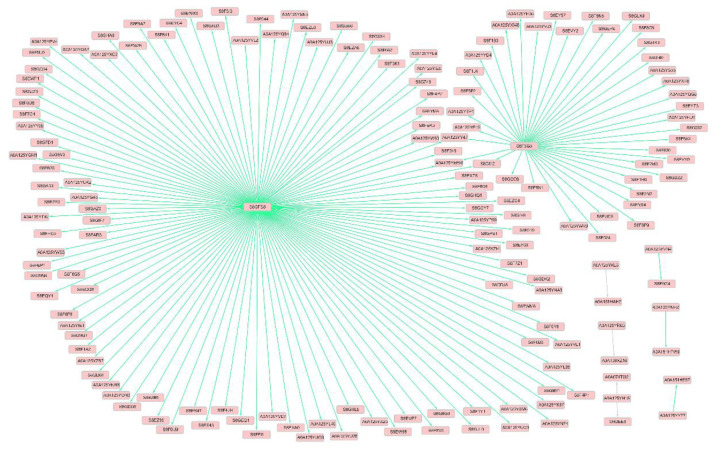
Network of kinase-phosphopeptide partners and potential phosphor- reliant interactions determined by correlation analysis. Light red nodes denoted phosphoproteins and edges represented interactors between phosphoproteins. The gray arrow denotes protein interactions which are presumed to be activated by the phosphorylation and green arrows represents protein interactions that are presumed to be inactivated by the phosphorylation.

**Table 1 molecules-27-01022-t001:** Recording of the kinase types corresponding to the motifs of upregulated phosphorylated peptides, as identified using KinasePhos.

Motif	Kinase Classes
P..SP	ATM, CDC2, CDK, CKI, CKII, IKK, MAPK, PKA, PKC, and PKG
E……SP	ATM, CaM-II, CDC2, CDK, CKI, CKII, IKK, INSR, MAPK, PKA, PKB, PKC, PKG, and Syk
SP…T	Ab1, ATM, CaM-II, CDC2, CDK, CKI, IKK, INSR, MAPK, PKA, PKB, PKC, and PKG
ASP	Ab1, ATM, CDC2, CDK, CKI, IKK, INSR, MAPK, PKA, PKB, PKC, PKG
K……SP	ATM, CDC2, CDK, CKI, CKII, IKK, MAPK, PKA, and PKG
SP….R	ATM, CaM-II, CDC2, CDK, CKI, CKII, IKK, Jak, MAPK, PKA, PKB, PKC, and PKG
P.SP	ATM, CaM-II, CDC2, CDK, CKI, IKK, MAPK, PKA, PKC, and PKG
SP	Ab1, ATM, CaM-II, CDC2, CDK, CKI, CKII, IKK, INSR, MAPK, Src, Syk, PKA, PKB, PKC, and PKG
L.R..S	ATM, CaM-II, CDC2, CDK, CKI, CKII, IKK, MAPK, PKA, PKB, PKC, and PKG
R..S	ATM, CaM-II, CDC2, CKI, CKII, IKK, MAPK, PKA, PKB, PKC, and PKG
GS	ATM, CaM-II, CDC2, CDK, CKI, CKII, IKK, INSR, Jak, MAPK, PKA, PKC, PKG, Src, and Syk
K..S	ATM, CDC2, CDK, CKI, IKK, MAPK, PKA, PKC, PKG, and Syk
S…..K	ATM, CaM-II, CDK, CKI, CKII, IKK, INSR, Jak, MAPK, PKB, PKC, PKG, and Syk
TP	ATM, CaM-II, CDC2, CDK, CKI, CKII, EGFR, IKK, INSR, Jak, MAPK, PKA, PKB, PKC, PKG, and Syk
R..T	Ab1, CaM-II, CDC2, EGFR, IKK, MAPK, PKA, PKC, PKG, and Src
K..T	CDK, IKK, MAPK, PKA, PKC, and PKG

**Table 2 molecules-27-01022-t002:** Recording of the kinase types corresponding to the motifs of downregulated phosphorylated peptides, as identified using KinasePhos.

Motif	Kinase Classes
RS.SP	ATM, CaM-II, CDC2, CDK, CKI, CKII, IKK, MAPK, PKA, PKB, PKC, and PKG
SP…K	ATM, PKA, CDC2, CDK, IKK, MAPK, PKC, and PKG
A.SP	ATM, CaM-II, CDC2, CDK, CKI, CKII, IKK, MAPK, PKA, PKB, PKC, PKG, and Syk
P.SP	ATM, CaM-II, CDC2, CDK, CKI, CKII, IKK, MAPK, PKA, PKB, PKC, and PKG
Q..SP	ATM, CaM-II, CDC2, CDK, CKI, CKII, IKK, MAPK, PKA, PKB, PKC, and PKG
G..SP	Ab1, ATM, CDC2, CDK, CKI, CKII, IKK, INSR, Jak, MAPK, PKB, PKC, PKG, and Syk
E.SP	ATM, CaM-II, CDC2, CDK, CKI, CKII, IKK, INSR, MAPK, PKA, PKB, PKC, and PKG
SP	Ab1, ATM, CaM-II, CDC2, CDK, CKI, CKII, IKK, INSR, MAPK, PKA, PKB, PKC, PKG, Src, and Syk
SD.E	ATM, CaM-II, CDC2, CDK, CKI, CKII, IKK, MAPK, PKA, PKB, PKC, and PKG
L.R..S	ATM, CaM-II, CDC2,CKI, CKII, IKK, INSR, MAPK, PKA, PKB, PKC, and PKG
R..S	ATM, CaM-II, CDC2, CDK, CKI, CKII, IKK, INSR, MAPK, PKA, PKB, PKC, PKG, Src, and Syk
SS.D	ATM, CaM-II, CDC2, CDK, CKI, CKII, IKK, MAPK, PKA, PKC, and PKG
SG.E	ATM, CaM-II, CDC2, CDK, CKI, CKII, IKK, INSR, MAPK, PKA, PKB, PKC, PKG, and Syk
SE.E	ATM, CaM-II, CDC2, CDK, CKI, CKII, EGFR, IKK, INSR, MAPK, PKA, PKB, PKC, PKG, and Syk
SD	Ab1, ATM, CaM-II, CDC2, CDK, CKI, CKII, IKK, INSR, MAPK, PKA, PKB, PKC, PKG, and Syk
GS	ATM, CaM-II, CDC2, CDK, CKI, CKII, IKK, INSR, MAPK, PKA, PKB, PKC, PKG, Src, and Syk
K..S	ATM, CaM-II, CDC2, CDK, CKI, CKII, IKK, INSR, MAPK, PKA, PKB, PKC, and PKG
DS	ATM, CaM-II, CDC2, CDK, CKI, CKII, EGFR, IKK, INSR, Jak, MAPK, PKA, PKC, PKC, and PKG
S..S	ATM, CaM-II, CDC2, CDK, CKI, CKII, IKK, MAPK, PKA, PKB, PKC, and PKG
G..TP	ATM, CDC2, CDK, CKII, IKK, MAPK, PKC, and PKG,
TP	ATM, CDC2, CDK, CKI, CKII, IKK, MAPK, PKA, PKB, and PKG,

## Data Availability

The identifier of the ProteomeXchange Consortium depositing mass spectrometry data is PXD017032.

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
