# Peer review of "Comparative Phosphoproteomic Analysis of Sporulated Oocysts and Tachyzoites of Toxoplasma gondii Reveals Stage-Specific Patterns"

_molecules, 2022, doi:10.3390/molecules27031022_

Round 1
Reviewer 1 Report
In this manuscript Wang et al. present new and relevant information based on a comparative analysis of the phosphoproteomics of sporulated oocysts and tachyzoites of a strain of Toxoplasma gondii isolated in China. The results were obtained using a methodology that involves IBT labeling, phosphopeptides enrichment using TiO2, and LC-MS/MS analysis. They clearly show significant differences in the amount of phosphopeptides between the two developmental stages of the protozoan.
My major concern is related with the use of sporulated oocysts. I would suggest to make the comparative analysis in tachyzoites, bradizoites and sporozoites of the strain. Purified sporozoites, rather than sporoulated oocysts, should be used in order to be sure you are comparying two different developmental stages. Sporulated oocysts is a crude preparation with the presence of remnants of the oocyst wall and sporocyst content. This point should be incorporated to the introduction and discussion sections.
Author Response
Response: We thank the reviewer very much for favorable comments and constructive suggestions on our manuscript. Virulent strain of T. gondii was generally deemed to have lost its ability to form a cyst and genotype Chinese 1 (ToxoDB#9) T. gondii has a semblable virulence to type 1 RH (Bansal et al., 2021; Frenkel et al., 1976; Shwab et al., 2018; Liang et al., 2020). Although it was reported that the cyst of a virulent RH strain of T. gondii might be produced in mice treated with sulfadiazine, the cyst of type one strains cannot be obtained in non-immune and non-treated mice, which means conducting research associated with cyst of a virulent strain of T. gondii during the natural course of infection will still be unavailable (Bansal et al., 2021; Boothroyd et al., 2002). Pure cyst of virulent ToxoDB#9 (PYS) strain of T. gondii was not successfully obtained either in mice or in guinea pigs. So bradyzoites was not included in life-cycle stages of virulent ToxoDB#9 (PYS) strain of T. gondii in comparative analysis. It is inevitable that pure sporozoite fractions were contaminated with oocyst wall fractions duo to sporozoite fractions deposited in oocyst walls in a continuous manner according to a previous study on the proteome of sporozoite and oocyst wall of T. gondii (Fritz et al., 2012). It seems that obtainment of pure sporozoites is unavailable. The critical structure of the oocyst wall of T. gondii provides protection to the sporozoites until its release, and research involving the structure and molecules of oocyst wall could support the detection and elimination of T. gondii oocysts in various environmental matrices (Freppel et al., 2018; Dumètre et al., 2013; Fritz et al., 2012). Therefore, purified sporoulated oocysts, rather than sporozoites, were used to make the comparative analysis in our study. We have incorporated these points to the introduction and discussion sections strictly according to the reviewer’s comments and suggestions.
Reviewer 2 Report
The manuscript “Comparative phosphoproteomic analysis of sporulated oocysts and tachyzoites of Toxoplasma gondii reveals stage-specific patterns” describes how phosphoproteomic was used to analyze the phosphorylation patter in tachyzoites and sporozoites of T. gondii strain Chinese1.
The primary concern about this manuscript is that the only data presented is the identification of phosphorylated peptides in both developmental stages of the parasite and further bioinformatics analysis. However, no secondary experiment or hypothesis is presented. The goal of this manuscript is unclear, and the experimental design is flawed as it only relies on one observation. Why did the authors decide to compare sporozoites to tachyzoites? What does the data reveal about sporozoite differentiation? Phosphorylation by itself has no biological meaning. Without an experimental analysis identifying the overall role of phosphorylation in the differentiation of sporozoites, or the characterization of one protein, this becomes meaningless. I would recommend, for example, the authors delete the SNf1 gene and characterize its differentiation phenotype.
I believe this dataset would be of incredible value if the authors design further experiments to validate it experimentally. But by itself isn’t enough for a publication, so I am rejecting this manuscript.
Minor reviews: the title of the manuscript has a typo.
Author Response
Response: We thank the reviewer very much for constructive suggestions on our manuscript. The differences of phosphoproteomes between different phases of Toxoplasma gondii remains a mystery. The goal of this manuscript are identification and analysis of the difference of phosphoproteomes between sporulated oocysts and tachyzoites of virulent ToxoDB#9 (PYS) strain of T. gondii. The sporozoites of T. gondii excysted from the oocyst and sporocyst walls invaded enteroepithelial cells of the intestine of the host followed by differentiation of tachyzoites (Freppel et al., 2018). Virulent strain of T. gondii was generally deemed to have lost its ability to form a cyst and genotype Chinese 1 (ToxoDB#9) T. gondii has a semblable virulence to type 1 RH (Bansal et al., 2021; Frenkel et al., 1976; Shwab et al., 2018; Liang et al., 2020). Although it was reported that cyst of a virulent RH strain of T. gondii might be produced in mice treated with sulfadiazine, the cyst of type one strains cannot be obtained in non-immune and non-treated mice, which means conducting research associated with the cyst of virulent strain of T. gondii during the natural course of infection will still be unavailable (Bansal et al., 2021; Boothroyd et al., 2002). Therefore, we identified and analyzed the difference of phosphoproteomes between sporulated oocysts and tachyzoites. It is inevitable that pure sporozoite fractions were contaminated with oocyst wall fractions duo to sporozoite fractions deposited in oocyst walls in a continuous manner according to a previous study on the proteome of sporozoite and oocyst wall of T. gondii (Fritz et al., 2012). It seems that obtainment of pure sporozoites is unavailable. Therefore, profile of phosphoproteins in pure sporozoites of virulent ToxoDB#9 (PYS) strain of T. gondii and role of phosphoproteins in the differentiation of sporozoites were not identified our study. We would like to delete the SNf1 gene and characterize its differentiation phenotype to validate it in our future research. We have revised the manuscript strictly according to the reviewer’s comments and suggestions. We have done our best to address all comments and we sincerely hope that you find our MS revised to your satisfaction.
Reviewer 3 Report
The article is focus on an interesting and important subject. I have only one remarks. The study should be complemented with the analysis of bradyzoites-containing cysts.
Author Response
Response: We thank the reviewer very much for favorable comments and constructive suggestions on our manuscript. Virulent strain of T. gondii was generally deemed to have lost its ability to form a cyst and genotype Chinese 1 (ToxoDB#9) T. gondii has a semblable virulence to type 1 RH (Bansal et al., 2021; Frenkel et al., 1976; Shwab et al., 2018; Liang et al., 2020). Although it was reported that the cyst of a virulent RH strain of T. gondii might be produced in mice treated with sulfadiazine, the cyst of type one strains cannot be obtained in non-immune and non-treated mice, which means conducting research associated with the cyst of virulent strain of T. gondii during the natural course of infection will still be unavailable (Bansal et al., 2021; Boothroyd et al., 2002). Pure cyst of virulent ToxoDB#9 (PYS) strain of T. gondii were not successfully obtained either in mice or in guinea pigs. So, bradyzoites-containing cysts was not included in life-cycle stages of virulent ToxoDB#9 (PYS) strain of T. gondii in our study. We have revised the manuscript strictly according to the reviewer’s comments and suggestions. We have done our best to address all comments and we sincerely hope that you find our MS revised to your satisfaction.
Round 2
Reviewer 3 Report
Dear Authors,
I thank and accept the justification.